# ATP hydrolysis by KaiC promotes its KaiA binding in the cyanobacterial circadian clock system

Yasuhiro Yunoki[1,3], Kentaro Ishii[1,2,3], Maho Yagi-Utsumi[1,2,3], Reiko Murakami[1], Susumu Uchiyama[2,4], Hirokazu Yagi[1], Koichi Kato[1,2,3]

The cyanobacterial clock is controlled via the interplay among KaiA, KaiB, and KaiC, which generate a periodic oscillation of KaiC phosphorylation in the presence of ATP. KaiC forms a homohexamer harboring 12 ATP-binding sites and exerts ATPase activities associated with its autophosphorylation and dephosphorylation. The KaiC nucleotide state is a determining factor of the KaiB–KaiC interaction; however, its relationship with the KaiA–KaiC interaction has not yet been elucidated. With the attempt to address this, our native mass spectrometric analyses indicated that ATP hydrolysis in the KaiC hexamer promotes its interaction with KaiA. Furthermore, our nuclear magnetic resonance spectral data revealed that ATP hydrolysis is coupled with conformational changes in the flexible C-terminal segments of KaiC, which carry KaiA-binding sites. From these data, we conclude that ATP hydrolysis in KaiC is coupled with the exposure of its C-terminal KaiA-binding sites, resulting in its high affinity for KaiA. These findings provide mechanistic insights into the ATP-mediated circadian periodicity.

## Introduction

Endogenous, entrainable oscillation with periods of ~24 h, known as the circadian rhythm, is found in many organisms. In cyanobacteria, the circadian rhythm is regulated by three clock proteins (Nakajima et al, 2005) (i.e., KaiA, KaiB, and KaiC) that autonomously undergo periodic assembly and disassembly irrespective of transcriptional and translational feedback systems (Tomita et al, 2005; Rust et al, 2007). KaiC is an AAA+ ATPase consisting of CI and CII domains and forms a hexameric ring structure, which harbors six ATP-binding sites in the CI ring and another six ATP-binding sites in the CII ring (Pattanayek et al, 2004). The KaiC hexamer experiences autophosphorylation and dephosphorylation cycles in a 24-h period through interactions with KaiA and KaiB in the presence of ATP

(Nakajima et al, 2005). Two phosphorylation sites at Ser431 and Thr432 in the KaiC CII domain (simply denoted as S and T, respectively) go through a cycle as follows: S/T → S/pT → pS/pT → pS/T → S/T, where "p" represents the phosphorylated residue (Nishiwaki et al, 2004). Phosphorylation oscillation is controlled via the interplay among KaiA, KaiB, and KaiC. KaiA and KaiB interact with the CII and CI domains of KaiC, respectively, so that its phosphorylation is up-regulated/down-regulated. Dephosphorylated KaiC interacts with KaiA, followed by an increase in the KaiC phosphorylation (Iwasaki et al, 2002; Williams et al, 2002). By contrast, KaiB interacts with phosphorylated KaiC and, thereby, accelerates its dephosphorylation (Kitayama et al, 2003; Xu et al, 2003).

A series of phospho-mimicking KaiC mutants have been widely used for characterizing the Kai protein complex formation in a phosphorylation state–dependent manner (Pattanayek et al, 2009, 2011; Lin et al, 2014; Sugiyama et al, 2016; Tseng et al, 2017; Mori et al, 2018). In these studies, it has been suggested that the interaction of KaiC with KaiB depends not only on the KaiC phosphorylation states but also on the nucleotide states in KaiC. For example, a phosphorylation-mimicking KaiC mutant was reactive to KaiB in the presence of ATP but loses the KaiB-binding affinity when adenylyl imidodiphosphate AMPPNP, a nonhydrolyzable ATP analog, was used instead of ATP (Phong et al, 2013; Mukaiyama et al, 2018). However, little is known about the relationship between the nucleotide state of KaiC and its KaiA-binding activity. Herein, we address this issue through characterizing ATP hydrolysis dependence of the KaiA–KaiC interaction by native mass spectrometry (MS), providing mechanistic insights into the binding of KaiA to the C-terminal segment of KaiC based on nuclear magnetic resonance (NMR) data.

## Results and Discussion

### KaiC ATP hydrolysis promotes KaiA–KaiC interaction

The phosphorylation states and the bound nucleotide states of KaiC are both governed by its ATP hydrolysis activity (Nishiwaki &

[1]Graduate School of Pharmaceutical Sciences, Nagoya City University, Nagoya, Japan   [2]Exploratory Research Center on Life and Living Systems (ExCELLS), National Institutes of Natural Sciences, Okazaki, Japan   [3]Institute for Molecular Science, National Institutes of Natural Sciences, Okazaki, Japan   [4]Department of Biotechnology, Graduate School of Engineering, Osaka University, Osaka, Japan

Correspondence: hyagi@phar.nagoya-cu.ac.jp; kkato@excells.orion.ac.jp
Reiko Murakami's Present address is Research Promotion and support headquarters, Fujita Health University Graduate School of Health Sciences, Toyoake, Aichi, Japan

Kondo, 2012; Nishiwaki-Ohkawa et al, 2014). To control the phosphorylation states, we used two KaiC mutants, KaiC$_{DD}$ and KaiC$_{AA}$ (in which Ser431 and Thr432 were both substituted with aspartate and alanine residues, respectively), mimicking the phosphorylated and dephosphorylated states of KaiC, respectively. It has been reported that KaiC$_{AA}$ shows enhanced complex formation with KaiA in comparison with KaiC$_{DD}$ (Lin et al, 2014; Tseng et al, 2014). To examine the possible dependence of KaiA–KaiC interaction on KaiC-bound nucleotide states, ATP and its nonhydrolyzable analog, adenylyl imidodiphosphate (AMPPNP), were used for the formation of the mutated KaiC hexamers, which were subjected to native MS analysis.

The native MS data confirmed that both KaiC$_{AA}$ and KaiC$_{DD}$ formed hexamers upon the addition of either ATP or AMPPNP (Figs 1A and B, S1A, and B). Whereas the KaiC$_{AA}$ hexamer mediated by AMPPNP had a constant molecular mass of 353,857 ± 10 D holding 12 AMPPNP molecules (Fig 1A and Table 1), the ATP-mediated KaiC$_{AA}$ hexamer preincubated for 5 h in the presence of 1 mM ATP exhibited two series of ion peaks (Fig 1B). The major peaks corresponded to the KaiC$_{AA}$ hexamer containing seven ATP and five ADP molecules, whereas the minor peaks corresponded to the KaiC$_{AA}$ hexamer containing seven ATP and three ADP molecules (Table 2). Native MS data of AMPPNP- or ATP-mediated KaiC$_{AA}$ hexamer incubated for 5 h in the presence of ATP and ADP at varying ratios, indicated that the nucleotide states of KaiC$_{AA}$ did not depend on the external ATP/ADP condition (Fig S2). These data indicate that the KaiC$_{AA}$ hexamer mediated by ATP spontaneously became asymmetric in

terms of nucleotide state and that the resultant ADP molecules were releasable. By contrast, the KaiC$_{DD}$ hexamer held at least 11 nucleotides, suggesting its low ATPase activity, which is consistent with a previous report that the ATPase activity of KaiC$_{AA}$ is six times higher than that of KaiC$_{DD}$ (Mutoh et al, 2013). Furthermore, we conducted tryptic fragmentation into CI and CII of the KaiC$_{AA}$ hexamer, followed by native MS analysis. The results revealed that the CI hexamer retained six nucleotides as prehydrolyzed ATP molecules, suggesting that the observed ATP hydrolysis exclusively occurred in the CII ring (Fig S3). This was supported by the observation that a KaiC$_{AA}$ mutant in which a catalytic glutamate (Glu77) in the CI ATPase domain (designated as KaiC$_{AA/E77Q}$) was substituted by glutamine exhibited virtually identical properties in terms of the nucleotide state and the KaiA binding to those of the KaiC$_{AA}$ hexamer (Fig S4 and Table 2).

Upon the addition of KaiA, both KaiC$_{AA}$ and KaiC$_{DD}$ formed ~420-kD complexes, indicating that two KaiA molecules bind one KaiC hexamer (Fig 1C and D, and Tables 1 and 2, and Fig S1C and D). This stoichiometry is consistent with the previously reported small-angle X-ray scattering and electron microscopy data, indicating that the KaiA dimer is tethered to the KaiC hexamer through its flexible C-terminal region (Pattanayek et al, 2006, 2011). Even in the presence of excess amounts of KaiA, the KaiA–KaiC complex were formed primarily in a 2:6 stoichiometry and in a 4:6 stoichiometry as minor complex (Fig S5). The occurrence of the complex significantly depended on the nucleotide state of KaiC and its phosphorylation mutation. The ATP-mediated KaiC$_{AA}$ hexamer almost exclusively

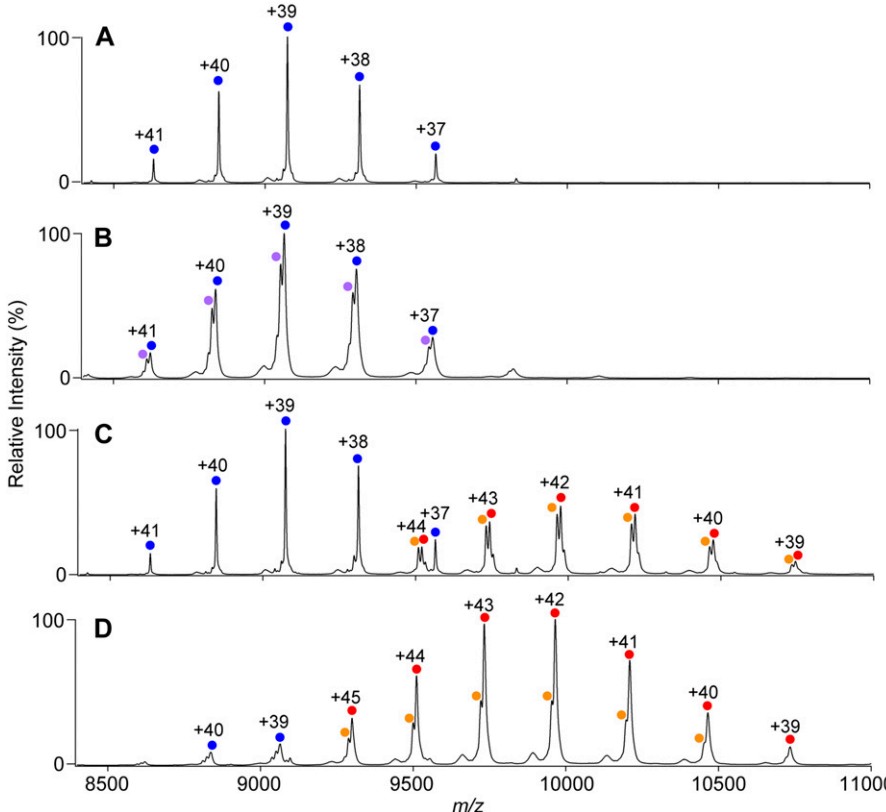

**Figure 1. KaiA–KaiC interaction depends on ATP hydrolysis.**
**(A–D)** Native mass spectra of (A, B) KaiC$_{AA}$ and (C, D) 6:3 mixtures of KaiC$_{AA}$ and KaiA in the presence of (A, C) 1 mM AMPPNP or (B, D) 1 mM ATP. After 5 h of incubation at 37°C with ATP or AMPPNP, the KaiC solutions with or without KaiA were immediately analyzed by nanoflow electrospray ionization MS. The blue and purple circles show the ion series of the KaiC$_{AA}$ homohexamer, whereas the orange and red circles show the 2:6 KaiA–KaiC$_{AA}$ hetero-octamer complexes. See Tables 1 and 2 for assignment details.

**Table 1. Summary of native MS characterization of KaiC and KaiA–KaiC complex formed in the presence of AMPPNP.**

| Figure number | Ion series | Assignment | | | Theoretical mass (D) | Experimental mass (D) | Δm (D)[a] | Relative quantity (%) |
|---|---|---|---|---|---|---|---|---|
| | | Protein complex | AMPPNP number | Mg$^{2+}$ number | | | | |
| Fig 1A | Blue | KaiC$_{AA6}$ | 12 | 12 | 353,850 | 353,857 ± 10 | −7 | — |
| Fig 1C | Blue | KaiC$_{AA6}$ | 12 | 12 | 353,850 | 353,855 ± 9 | −5 | — |
| Fig 1C | Red | KaiC$_{AA6}$/KaiA$_2$ | 11 | 12 | 418,838 | 418,896 ± 27 | −58 | 53[b] |
| Fig 1C | Orange | KaiC$_{AA6}$/KaiA$_2$ | 10 | 12 | 418,332 | 418,412 ± 28 | −80 | 47[b] |
| Fig S2A | Blue | KaiC$_{DD6}$ | 12 | 12 | 354,396 | 354,375 ± 12 | 21 | — |
| Fig S2C | Blue | KaiC$_{DD6}$ | 12 | 12 | 354,396 | 354,447 ± 16 | −51 | — |
| Fig S2C | Red | KaiC$_{DD6}$/KaiA$_2$ | 12 | 12 | 419,890 | 420,076 ± 47 | −186 | — |

[a]Δm is the mass difference between the experimental mass and the theoretical mass.
[b]Relative quantity of two ion series are shown.

formed the complex with KaiA, whereas the complex formation was compromised in the AMPPNP-mediated KaiC$_{AA}$ hexamer and both the AMPPNP- and ATP-mediated KaiC$_{DD}$ hexamers. These data indicate that the nonphosphorylated KaiC hexamer becomes most reactive with KaiA after ATP hydrolysis. Interestingly, in most cases, the KaiA–KaiC complexes lost one or two nucleotides (Tables 1 and 2). Noteworthy, the AMPPNP-mediated KaiC$_{AA}$ hexamer in complex with the KaiA dimer lacked one and two nucleotides, strongly suggesting that KaiA binding promotes nucleotide release from KaiC.

On the basis of our data and previous reports that KaiA binding stimulates the ATPase activity of KaiC (Terauchi et al, 2007; Murakami et al, 2008), we conclude that ATP hydrolysis of KaiC promotes its interaction with KaiA, which is coupled with ADP release and, in turn, enhances the KaiC ATPase activity.

### KaiC ATP hydrolysis triggers exposure of its KaiA-binding segments

Using NMR-based structural characterization, we attempted to address how ATP hydrolysis in KaiC affects its binding to KaiA. Because of the fact that KaiA was reported to interact with the A-loop (Glu487-Ile497) and C-tail (Ser498-Ser518) of KaiC (Vakonakis & LiWang, 2004; Pattanayek et al, 2006, 2011; Pattanayek & Egli, 2015), we focused on these C-terminal segments. We compared the $^1$H-$^{15}$N HSQC spectra of KaiC$_{AA}$ and its mutant with the deletion of the C-terminal region 487–518 in the presence of AMPPNP or ATP (Fig S6). Despite its high molecular mass (347 kD), the KaiC$_{AA}$ hexamer yielded ~30 observable peaks originating from the backbone amide groups with high flexibility, most of which disappeared in the spectrum of the truncated mutant and, therefore, were assigned to the C-terminal region. The signals derived from the C-terminal region were assigned to individual amino acid residues located in the A-loop and C-tail by a series of $^1$H/$^{13}$C/$^{15}$N multinuclear NMR experiments (Fig 2A and B). In the KaiC$_{AA}$ hexamer mediated by AMPPNP, these peaks demonstrated enhanced peak broadening in comparison with those from the ATP-mediated hexamer and, importantly, Gly488, Ile489, and Ile497 yielded no observable peak, indicating that its A-loop became mobile upon ATP hydrolysis (Fig 2D and F). Furthermore, the peaks originating from the C-terminal segments, including Gly488 and Ile489 in the A-loop,

were attenuated or even disappeared upon the addition of KaiA in the presence of ATP, indicating that both the A-loop and the C-tail are involved in the interaction with KaiA (Fig 2C and E). This result is consistent with the X-ray crystal structure, which shows that the full-length *Synechococcus elongatus* KaiA homodimer is in complex with two KaiC C-tail peptides (Asp500-Ser519) (Pattanayek & Egli, 2015) and also that the NMR solution structure of the C-terminal domain of *Thermosynechococcus elongatus* KaiA (residues 180–283) interacting with a C-terminal peptide includes the A-loop and C-tail (residues 488–518) of KaiC. In the latter case, the KaiC-derived peptide binds a KaiA surface through its eight hydrophobic side chains in the A-loop (Ile490, Thr493, and Thr495) and in the C-tail (Val499, Leu505, Ala506, Ile508, and Met512) (Vakonakis & LiWang, 2004).

To date, the crystal structures of the wild-type KaiC hexamer and its phosphorylated mutants have been solved as AMPPNP-bound states (Pattanayek et al, 2004, 2009, 2014). In these crystal structures, the A-loop forms a U-shaped hairpin, which is accommodated in a cleft proximal to the nucleotide-binding site and is followed by the C-tail, which is exposed to the solvent and gives no interpretable electron density for the segment beyond E504 (Fig 2F) (Pattanayek & Egli, 2015). Under this circumstance, the interaction of KaiA with the KaiC hexamer is compromised because of the inaccessibility of the A-loop. Our MS and NMR data indicate that, upon ATP hydrolysis, the A-loop is released and becomes reactive with KaiA, leading to the enhanced KaiA–KaiC interaction.

The crystal structure revealed that AMPPNP molecules are located at the intersubunit interfaces of the KaiC hexamer (Pattanayek et al, 2014). In the CII ring, a loop comprising residues 415–433 (termed "422-loop") makes contact with both AMPPNP and the A-loop (Fig 2F) (Vakonakis & LiWang, 2004; Kim et al, 2008; Egli et al, 2013). Based on our findings in conjunction with the crystallographic data, we propose a mechanistic model in which ATP hydrolysis in the CII ring triggers a conformational rearrangement of the 422-loop, resulting in "leaping out" of the A-loop so as to interact with KaiA. Reciprocally, it is possible that KaiA pulls out the A-loop and, thereby, causes a microenvironmental rearrangement surrounding the nucleotide-binding site of the KaiC CII ring, facilitating ADP release from it. This is supported by a recently reported molecular dynamics simulation of the KaiC hexamer (Hong et al, 2018).

**Table 2. Summary of native MS characterization of KaiC and KaiA–KaiC complex formed in the presence of ATP.**

| Figure number | Ion series | Assignment | | | | Theoretical mass (D) | Experimental mass (D) | $\Delta m$ (D)[a] | Relative quantity (%) |
| | | Protein complex | ATP number | ADP number | $Mg^{2+}$ number | | | | |
|---|---|---|---|---|---|---|---|---|---|
| Fig 1B | Blue | $KaiC_{AA6}$ | 7 | 5 | 12 | 353,462 | 353,476 ± 18 | −14 | 56[b] |
| Fig 1B | Purple | $KaiC_{AA6}$ | 7 | 3 | 12 | 352,608 | 352,593 ± 16 | 15 | 44[b] |
| Fig 1D | Blue | $KaiC_{AA6}$ | 7 | 5 | 12 | 353,462 | 353,461 ± 14 | 1 | — |
| Fig 1D | Red | $KaiC_{AA6}/KaiA_2$ | 6 | 5 | 12 | 418,449 | 418,445 ± 20 | 4 | 67[b] |
| Fig 1D | Orange | $KaiC_{AA6}/KaiA_2$ | 0 | 11 | 12 | 417,969 | 417,963 ± 20 | 6 | 33[b] |
| | | $KaiC_{AA6}/KaiA_2$ | 5 | 5 | 12 | 417,942 | 417,963 ± 20 | −21 | |
| Fig S1B | Blue | $KaiC_{DD6}$ | 11 | 0 | 12 | 353,901 | 353,902 ± 11 | −1 | — |
| | | $KaiC_{DD6}$ | 6 | 6 | 12 | 353,928 | 353,902 ± 11 | 26 | |
| Fig S1D | Blue | $KaiC_{DD6}$ | 6 | 6 | 12 | 353,928 | 353,924 ± 9 | 4 | 74[b] |
| | | $KaiC_{DD6}$ | 11 | 0 | 12 | 353,901 | 353,924 ± 9 | −23 | |
| Fig S1D | Purple | $KaiC_{DD6}$ | 6 | 5 | 12 | 353,501 | 353,477 ± 9 | 24 | 26[b] |
| | | $KaiC_{DD6}$ | 0 | 12 | 12 | 353,448 | 353,477 ± 9 | −29 | |
| Fig S1D | Red | $KaiC_{DD6}/KaiA_2$ | 2 | 10 | 12 | 419,102 | 419,102 ± 32 | 0 | — |
| | | $KaiC_{DD6}/KaiA_2$ | 7 | 4 | 12 | 419,075 | 419,102 ± 32 | −27 | |
| Fig S4A | Blue | $KaiC_{AA/E77Q6}$ | 7 | 5 | 12 | 353,396 | 353,383 ± 13 | 13 | — |
| Fig S4B | Blue | $KaiC_{AA/E77Q6}$ | 12 | 0 | 12 | 353,796 | 353,766 ± 29 | 30 | — |
| | Red | $KaiC_{AA/E77Q6}/KaiA_2$ | 7 | 4 | 12 | 418,533 | 418,559 ± 59 | −26 | — |

[a]$\Delta m$ is the mass difference between the experimental mass and the theoretical mass.
[b]Relative quantity of two ion series are shown.

The present study experimentally reveals that ATP hydrolysis in the KaiC hexamer triggers the exposure of its C-terminal segments into the solvent so as to capture KaiA, which in turn facilitates ADP release. These findings imply that the ATPase-dependent KaiA interaction promotes ADP/ATP turnover on KaiC, leading to the up-regulation of its autophosphorylation (Fig 3). Our findings provide mechanistic insights into the circadian periodicity mediated by the unique AAA+ ATPase.

# Materials and Methods

### Protein expression and purification for native MS

KaiA and KaiC mutants originating from thermophilic cyanobacteria, *T. elongatus* BP-1, were expressed in *Escherichia coli* and purified as previously described (Ishii et al, 2014). The expression plasmids of the KaiC mutants ($KaiC_{AA}$, $KaiC_{AA/E77Q}$, and $KaiC_{DD}$) were also constructed according to a previous study (Murakami et al, 2012; Mutoh et al, 2013).

### Protein expression and purification for NMR spectroscopy

The expression plasmids of $KaiC_{AA}$ and its mutant with the deletion of the C-terminal segment 487–518 were constructed according to a previous study (Ishii et al, 2014). For NMR analyses, the protein was expressed in M9 minimal medium containing $[^{15}N]NH_4Cl$ and/or $[^{13}C]$glucose. The protein was purified according to a previous report (Ishii et al, 2014). The NMR samples were prepared by dissolving the mutated KaiC (53 $\mu$M final concentration) in 10% $D_2O$ containing 20 mM Tris–HCl (pH 7.0), 150 mM NaCl, 5 mM $MgCl_2$, 0.5 mM EDTA, and 1 mM DTT in the presence of 1 mM ATP or AMPPNP.

### Native MS analyses

For KaiC hexamerization, the purified monomeric KaiC mutants ($KaiC_{AA}$, $KaiC_{AA/E77Q}$, and $KaiC_{DD}$) (20 $\mu$M) were incubated in the presence of 1 mM nucleotide (ATP or AMPPNP) for 30 min on ice. The KaiC solutions were incubated at 37°C for 5 h in the presence of ATP or AMPPNP. In the examination of possible effects of the external ATP/ADP variation, the AMPPNP- or ATP-mediated $KaiC_{AA}$ hexamer was incubated in the presence of ATP and ADP at varying ratios. Subsequently, these KaiC solutions, in the absence or presence of KaiA (10 $\mu$M), were buffer-exchanged into 150 mM ammonium acetate (pH 6.8) by passing the proteins through a Bio-Spin 6 column (Bio-Rad) and were then immediately subjected to nanoflow electrospray ionization MS using gold-coated glass capillaries made in-house (~2–5 $\mu$l sample loaded per analysis). For the tryptic fragmentation into CI and CII of the $KaiC_{AA}$ hexamer, $KaiC_{AA}$ incubated at 37°C in the presence of 1 mM ATP for 5 h was buffer-exchanged into 150 mM aqueous ammonium acetate and then digested with 0.02 mg/ml trypsin for up to 60 min, followed directly by nanoflow electrospray ionization MS measurements. Spectra

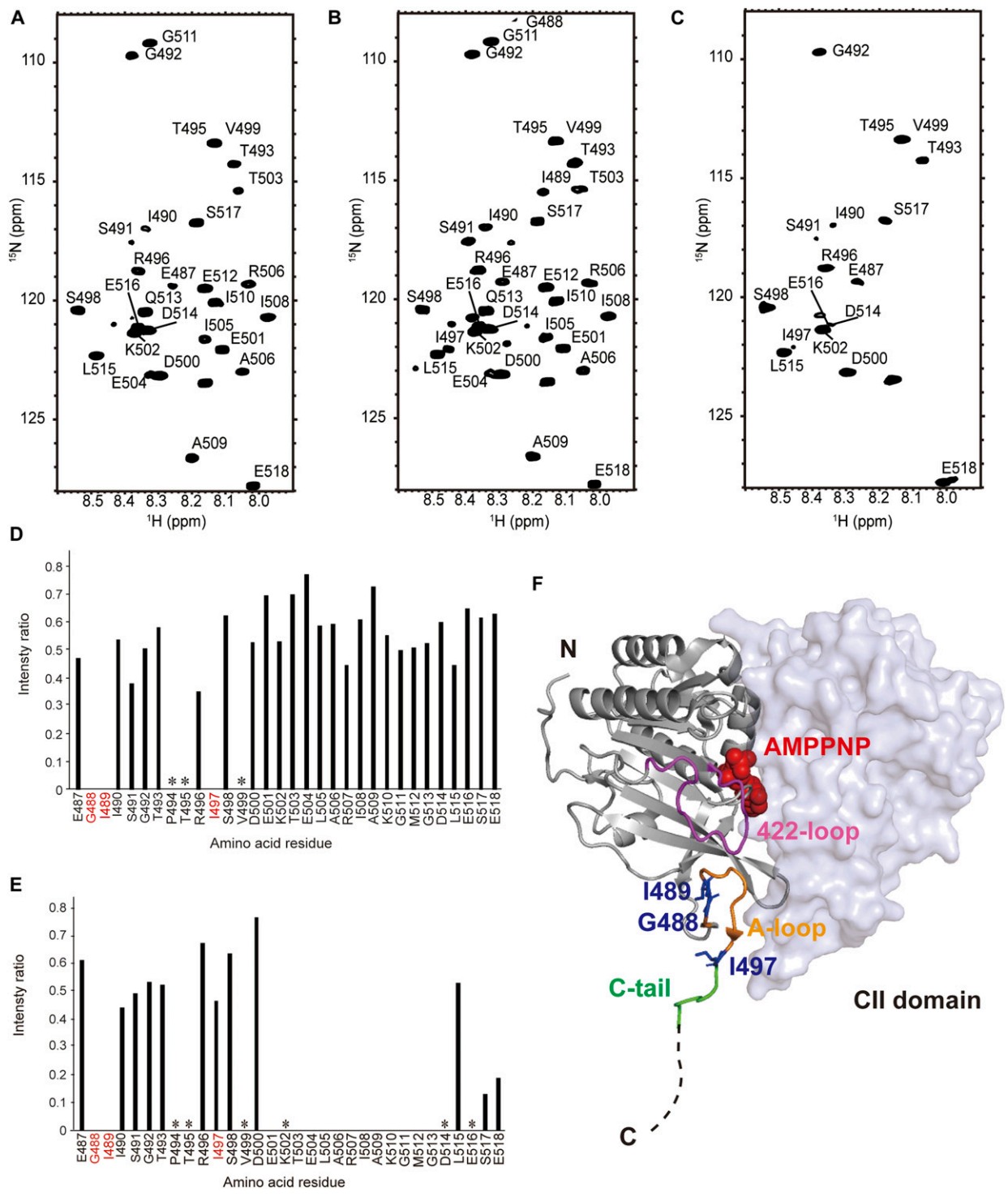

**Figure 2. ATP hydrolysis–dependent conformational change of the C-terminal KaiA-binding region of KaiC.**

**(A–C)** $^{1}$H-$^{15}$N HSQC spectrum of KaiC$_{AA}$ in the presence of (A) AMPPNP, (B) ATP, and (C) KaiA and ATP. NMR experiments were set up to take a total time of 3 h using the KaiC hexamer incubated with AMPPNP or ATP for 9 h. Assignments of the peaks from the C-terminal region are given in each spectrum. **(D)** Plot of relative peak intensity for KaiC$_{AA}$ resonances in the presence of AMPPNP versus ATP. **(E)** Plot of relative peak intensity for KaiC$_{AA}$ resonances in the presence versus absence of KaiA under the ATP condition. In (D) and (E), the residues that yielded no observable peaks under the AMPPNP condition are highlighted in red, whereas the asterisks indicate the proline residues and residues whose chemical shift perturbation data could not be obtained because of severe peak overlapping. **(F)** Crystal structure of two KaiC protomers in cartoon and surface representation, respectively, in the KaiC homohexameric ring mediated by AMPPNP (PDB ID code: 4OOM). In the crystal structure, the C-terminal region comprises a U-shaped A-loop (Glu487-Ile497) (orange) and a solvent-exposed C-tail (S498-S518), in which only the Ser498-Glu504 part (green) was modeled. The three residues (i.e., Gly488, Ile489, and Ile497) located in the A-loop, whose HSQC peaks were unobserved under the AMPPNP condition, are colored blue. The A-loop and AMPPNP molecule (red) are mediated by a loop comprising residues 415–430 (termed 422-loop, magenta).

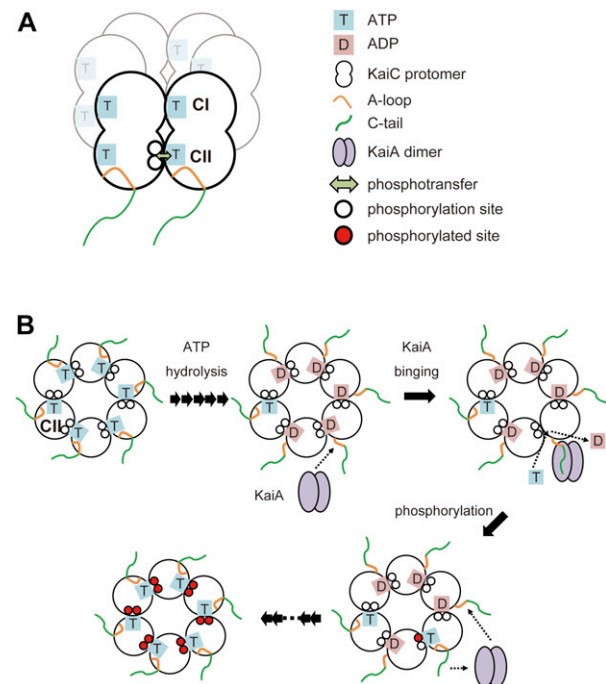

**Figure 3. The "fishing a line" mechanism coupling ATP hydrolysis and KaiA-mediated up-regulation of autophosphorylation in the KaiC hexamer.**
**(A)** While both CI and CII domains harbor nucleotide-binding sites and ATPase-active sites at the subunit interfaces, the autokinase activity is exerted only in the CII domain. This is because the autophosphorylation sites (i.e., Ser431 and Thr432) are spatially proximal to the ATP molecule accommodated in the CII domain of the neighboring protomer. **(B)** In the CII AAA+ ring hexamer, ATP hydrolysis releases the A-loop, which thereby becomes reactive with KaiA. KaiA binding to the C-terminal segments of KaiC facilitates ADP release and ATP incorporation. The rapid ATP/ADP turnover leads to the up-regulation of autophosphorylation of KaiC.

were recorded on a SYNAPT G2-S*i* HDMS mass spectrometer (Waters) in positive ionization mode at 1.33 kV with a 150 V sampling cone voltage and source offset voltage, 0 V trap and transfer collision energy, and 5 ml/min trap gas flow. Spectra were calibrated using 1 mg/ml cesium iodide and analyzed using MassLynx software (Waters) (Sugiyama et al, 2016).

### NMR analyses

All NMR spectra were acquired at 37°C using Bruker DMX-500 and Bruker AVANCE 800 US spectrometers. The chemical shifts of KaiC peaks were assigned on the basis of 2D $^1$H-$^{15}$N HSQC, 3D $^1$H-$^{13}$C-$^{15}$N HSQC, 3D HNCO, and 3D HNCA spectral data in the presence of AMPPNP or ATP. The purified monomeric KaiC with isotope labeling (53 $\mu$M) was buffer-exchanged into 20 mM Tris buffer (pH 7.0) containing 150 mM NaCl, 5 mM MgCl$_2$, 0.5 mM EDTA, 1 mM nucleotide (ATP or AMPPNP), and 10% (vol/vol) D$_2$O and was then incubated at 4°C for 30 min for hexamerization. The KaiC hexamers formed by AMPPNP were used for the NMR assignment experiment. After 9 h of incubation at 37°C, the KaiC solutions were subjected to $^1$H-$^{15}$N HSQC measurements in the absence and presence of a KaiA dimer (53 $\mu$M). All NMR data were processed using NMRPipe software and analyzed with Sparky software (Delaglio et al, 1995; Lee et al, 2009).

## Supplementary Information

## Acknowledgements

We thank Dr. T Suzuki (ExCELLS) for his assistance in MS analysis. This work was supported by grants (JP18J21063 to Y Yunoki, JP15K18492 to R Murakami, JP18K14671 to K Ishii, and JP25102001 and JP25102008 to K Kato) from the Ministry of Education, Culture, Sports, Science and Technology (MEXT) of Japan, by the Okazaki ORION project, and by the Joint Research by ExCELLS. We acknowledge the assistance of the Research Equipment Sharing Center at the Nagoya City University.

### Author Contributions

Y Yunoki: conceptualization, data curation, formal analysis, funding acquisition, investigation, visualization, methodology, and writing—original draft.
K Ishii: data curation, formal analysis, validation, investigation, and methodology.
M Yagi-Utsumi: data curation, formal analysis, investigation, methodology, and project administration
R Murakami: investigation.
S Uchiyama: supervision and project administration.
H Yagi: conceptualization, formal analysis, supervision, visualization, project administration, and writing—original draft, review, and editing.
K Kato: conceptualization, resources, supervision, funding acquisition, project administration, and writing—review and editing.

### Conflict of Interest Statement

The authors declare no conflict of interest.

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
