## [Reviewer comments · Life Science Alliance]

Life Science Alliance

ATP Hydrolysis by KaiC Promotes Its KaiA Binding in the Cyanobacterial Circadian Clock System

Yasuhiro Yunoki, Kentaro Ishii, Maho Yagi-Utsumi, Reiko Murakami, Susumu Uchiyama, Hirokazu Yagi, and Koichi Kato

DOI: <https://doi.org/10.26508/lsa.201900368>

Corresponding author(s): Koichi Kato, National Institutes of Natural Sciences and Hirokazu Yagi, Nagoya City University

Review Timeline:

Submission Date:	2019-03-06
Editorial Decision:	2019-03-18
Revision Received:	2019-05-01
Editorial Decision:	2019-05-13
Revision Received:	2019-05-16
Accepted:	2019-05-17

Scientific Editor: Andrea Leibfried

Transaction Report:

March 18, 2019

Re: Life Science Alliance manuscript #LSA-2019-00368-T

Koichi Kato
National Institutes of Natural Sciences
Exploratory Research Center on Life and Living Systems (ExCELLS)

Dear Dr. Kato,

Thank you for submitting your manuscript entitled "ATP Hydrolysis by KaiC Promotes Its KaiA Binding in the Cyanobacterial Circadian Clock System" to Life Science Alliance. The manuscript was assessed by expert reviewers, whose comments are appended to this letter.

As you will see, the reviewers appreciate your data but think that your conclusions are not sufficiently supported by the data currently provided. They provide constructive input on how to strengthen your work, and we would thus like to invite you to submit a revised version to us. Importantly, controls (mutant analysis), additional time points and experiments in more physiological conditions should get added as well as interpretations revisited.

Thank you for this interesting contribution to Life Science Alliance. We are looking forward to receiving your revised manuscript.

Sincerely,

Andrea Leibfried, PhD
Executive Editor

Life Science Alliance
Meyerhofstr. 1
69117 Heidelberg, Germany
t +49 6221 8891 502
e a.leibfried@life-science-alliance.org
www.life-science-alliance.org

B. MANUSCRIPT ORGANIZATION AND FORMATTING:

Reviewer #1 (Comments to the Authors (Required)):

Review of Yunoki, Y. et al., "ATP hydrolysis by KaiC promotes its KaiA binding in the cyanobacterial circadian clock system"

The cyanobacterial clock is an elegant biochemical timer that measures out ~24-hour increments based on post-translational changes in three clock proteins, KaiA, B and C in response to ATP hydrolysis and phosphorylation on KaiC. While many studies have looked at KaiA binding to the C-

terminal regions of KaiC, there are still major questions to resolve, such as identifying how the accessibility of these flexible regions is regulated to control KaiA binding and subsequent regulation of KaiC phosphorylation state. This manuscript addresses the relationship between the KaiC nucleotide state (i.e. ATP hydrolysis) and its ability to bind KaiA using native mass spectrometry (MS) and NMR approaches. The native MS data provide unprecedented insight into the nucleotide binding state of the KaiC hexamer revealing that ATP hydrolysis is a critical step in expelling the A-loops for KaiA binding. The clever use of NMR in this large system allows for monitoring of the flexible A-loops and C-terminal tails in the context of KaiC and chemical shift assignment of the peaks and the analysis of their binding is a valuable contribution to the field. Overall, the data here show that ATP hydrolysis by KaiC allows the A-loop to be released and facilitate binding to KaiA, which then stimulates release of ADP so it can rebind ATP and phosphorylate either S431 or T432. This is an interesting model that has mechanistically important implications for the clock. The domain responsible for this critical ATP hydrolytic event is not stated explicitly in the text, although it is depicted that the ATP hydrolysis driving release of the A-loops occurs in the CII domain in Fig. 2. While this assumption seems reasonable, it was not demonstrated conclusively. The conclusion here might be strengthened by performing these experiments in the presence of a KaiC mutant with catalytically inactive CI domain, allowing the authors to assign this mechanistically important step to the CII domain.

Major comments:

1) It is assumed here that the ATP hydrolysis necessary to release the A-loops occurs in the KaiC CII domain, but no controls are provided in support of this. Could this be tested by looking at ATP hydrolysis and KaiA binding in a KaiC mutant with a catalytically dead CI domain (E77Q/E78Q)?

Minor comments:

- 1) The phrase "Even in the phosphorylation-mimicking KaiC mutant..." at the bottom of page 2 could be rephrased for better clarity.
- 2) Given that you define the stoichiometry of KaiA binding to KaiC as 2 dimers per hexamer (page 3), it would be nice if it were made clear in the main text that an excess of KaiA was added.
- 3) The description of KaiA binding to the KaiCAA and DD state is a bit confusing - it is not explicitly stated that ATP-mediated hexamers readily formed complexes; later in the same paragraph, we're told that complex formation was compromised in the AMPPNP-mediated hexamers, so I'm guessing this is the case.
- 4) While the NMR data are quite beautiful, some of the terminology used to describe it is not quite standard. Instead of saying "these peaks exhibited intensity attenuation" (middle of page 4), it might be better to use the more common phrase "these peaks demonstrated enhanced peak broadening".
- 5) It is a bit hard to see the cyan lettering in Fig. 2 panel f.
- 6) Cartoons of KaiC are almost always drawn with the CII domain on top. The cartoon in Fig. 3 should either be re-oriented so the A-loops are on the top, or the CI and CII domains should be labeled here.

Reviewer #2 (Comments to the Authors (Required)):

The authors analysed here the interaction between the KaiC ATPase and the regulatory subunit KaiA. Briefly, the protein interaction between both proteins is an important step for the generation

of about 24 h (circadian) rhythms in cyanobacteria. Based on native mass spectrometry and NMR data a model for the interaction was formulated. The formation of ADP from ATP releases part of the C-terminus of KaiC to act as anchor for a KaiA homodimer, which facilitates the release of ADP. Taken together, the experiments performed and the model drawn are a significant progress for the field.

However, I had two questions:

The methods are restricted to one point. Is it therefore feasible to draw a dynamic model? Would it be possible to measure kinetics to monitor binding of KaiA and the kinetics of ADP release before and after? At least the peaks with different ADP content should be quantified.

By comparison of the intensity ratios between figures 5D and 5E it is apparent that the changes occur starting from I497, which is at the border between the A-loop and the C-tail. Hence, is it really possible to state that the A-loop becomes exposed by the interaction with KaiA? It looks like only the C-tail is involved.

Minor points:

The figure panel meant in the phrase 'ion peaks (Fig 1C)' is probably Fig 1B.

Figure legend 1: replace '1 : 1 mixtures' by '1 : 2 mixtures' (or mention KaiA homodimers).

Reviewer #3 (Comments to the Authors (Required)):

In this clearly written manuscript by Yunoki and coworkers, "ATP hydrolysis by KaiC promotes its KaiA binding in the cyanobacterial circadian clock system", data is presented demonstrating that ATP hydrolysis enhances the KaiA-KaiC interaction.

The idea that ATP hydrolysis in the C2 ring of KaiC is a prerequisite for KaiA binding is a new contribution to the field, and the data presented here support the authors' contention. The MS and NMR data are of high quality and were interpreted correctly.

Until now, it was thought that the C2 phosphorylation state of KaiC solely dictated the dynamic in/out equilibrium of the A-loops. Whether ADP or ATP were bound by the C2 ring was never considered. Here, data suggest that A-loop exposure requires that ADP occupy at least a fraction of the six nucleotide-binding sites in the C2 ring.

There is one caveat this reviewer would like to point out. The experiments involving ATP were non-physiological in the sense that reaction mixtures contained 100% ATP, rather than more physiological conditions, like an 80%/20% ATP/ADP ratio. Under physiological conditions it is possible that the C2 ring would have ADP at some sites, but not as products of C2 ATP hydrolysis. Thus, it can be envisioned that under physiological conditions the in/out equilibrium of A-loops would be shifted sufficiently toward the out conformation even without a C2 ATPase. This question can be easily addressed by repeating experiments under a physiological range of ATP/ADP ratios. At the very least, however, the authors should discuss how using non-physiological 100% ATP could bias their results.

NMR peak intensity ratios in Fig. 2D & 2E: any differences in protein concentrations between samples would cause systematic errors. Surprisingly, Fig. 2E suggests that the peak intensity for residue K502 does not change significantly when KaiA binds KaiC. Authors should address how differences in protein concentrations between samples were considered/corrected/eliminated.

Reviewer #1 (Comments to the Authors (Required)):

Review of Yunoki, Y. et al., "ATP hydrolysis by KaiC promotes its KaiA binding in the cyanobacterial circadian clock system"

The cyanobacterial clock is an elegant biochemical timer that measures out ~24-hour increments based on post-translational changes in three clock proteins, KaiA, B and C in response to ATP hydrolysis and phosphorylation on KaiC. While many studies have looked at KaiA binding to the C-terminal regions of KaiC, there are still major questions to resolve, such as identifying how the accessibility of these flexible regions is regulated to control KaiA binding and subsequent regulation of KaiC phosphorylation state. This manuscript addresses the relationship between the KaiC nucleotide state (i.e. ATP hydrolysis) and its ability to bind KaiA using native mass spectrometry (MS) and NMR approaches. The native MS data provide unprecedented insight into the nucleotide binding state of the KaiC hexamer revealing that ATP hydrolysis is a critical step in expelling the A-loops for KaiA binding. The clever use of NMR in this large system allows for monitoring of the flexible A-loops and C-terminal tails in the context of KaiC and chemical shift assignment of the peaks and the analysis of their binding is a valuable contribution to the field. Overall, the data here show that ATP hydrolysis by KaiC allows the A-loop to be released and facilitate binding to KaiA, which then stimulates release of ADP so it can rebind ATP and phosphorylate either S₄₃₁ or T₄₃₂. This is an interesting model that has mechanistically important implications for the clock. The domain responsible for this critical ATP hydrolytic event is not stated explicitly in the text, although it is depicted that the ATP hydrolysis driving release of the A-loops occurs in the CII domain in Fig. 2. While this assumption seems reasonable, it was not demonstrated conclusively. The conclusion here might be strengthened by performing these experiments in the presence of a KaiC mutant with catalytically inactive CI domain, allowing the authors to assign this mechanistically important step to the CII domain.

Major comments:

1) *It is assumed here that the ATP hydrolysis necessary to release the A-loops occurs in the KaiC CII domain, but no controls are provided in support of this. Could this be tested by looking at ATP hydrolysis and KaiA binding in a KaiC mutant with a catalytically dead CI domain (E77Q/E78Q)?*

We thank the reviewer for the constructive comment. To address this issue, we examined KaiA

binding to a KaiC_{AA} mutant, KaiC_{AA/E77Q}. The native mass spectrometric (MS) data indicated that the mutation affected neither the nucleotide state nor the KaiA-binding property of the of the KaiC_{AA} hexamer. In addition, we performed tryptic fragmentation into CI and CII of the KaiC_{AA} hexamer preincubated for 5 h in the presence of ATP, followed by native MS analysis. The result revealed that the C1 hexamer retained six nucleotides as prehydrolyzed ATP molecules, indicating that the observed ATP hydrolysis exclusively occurred in the CII ring. In the revised manuscript, we added these new experimental results in Supplementary Figures S3 and S4 and modified the text accordingly (p.3, lines 25-31).

Minor comments:

1) The phrase "Even in the phosphorylation-mimicking KaiC mutant..." at the bottom of page 2 could be rephrased for better clarity.

According to the reviewer's comment, we described this mutant more clearly in the revised manuscript (p.2, lines 31-33).

2) Given that you define the stoichiometry of KaiA binding to KaiC as 2 dimers per hexamer (page 3), it would be nice if it were made clear in the main text that an excess of KaiA was added.

We appreciate this useful comment and conducted titrated KaiC_{AA} with KaiA. The result indicated that, even in the presence of excess amounts of KaiA, the KaiA-KaiC complex were formed primarily in a 2 : 6 stoichiometry and in a 4 : 6 stoichiometry as minor complex. We have now added a Supplementary Figure S5 and described the results in the main text in the revised manuscript (p.4, lines 1-2).

3) The description of KaiA binding to the KaiCAA and DD state is a bit confusing - it is not explicitly stated that ATP-mediated hexamers readily formed complexes; later in the same paragraph, we're told that complex formation was compromised in the AMPPNP-mediated hexamers, so I'm guessing this is the case.

We thank the reviewer for the helpful suggestion. We clarified the nucleotide state of KaiC_{DD} in the revised manuscript (p.4, line 5).

4) While the NMR data are quite beautiful, some of the terminology used to describe it is not quite standard. Instead of saying "these peaks exhibited intensity attenuation" (middle of page 4), it might be better to use the more common phrase "these peaks demonstrated enhanced peak broadening".

As per the reviewer's comment, we have rephrased the sentence (p.4, lines 24-25).

5) It is a bit hard to see the cyan lettering in Fig. 2 panel f.

According to the reviewer's comment, we used magenta color in Figure 2.

6) Cartoons of KaiC are almost always drawn with the CII domain on top. The cartoon in Fig. 3 should either be re-oriented so the A-loops are on the top, or the CI and CII domains should be labeled here.

As per the reviewer's comment, the CI and CII domains were labeled in Figure 3.

Reviewer #2 (Comments to the Authors (Required)):

The authors analysed here the interaction between the KaiC ATPase and the regulatory subunit KaiA. Briefly, the protein interaction between both proteins is an important step for the generation of about 24 h (circadian) rhythms in cyanobacteria. Based on native mass spectrometry and NMR data a model for the interaction was formulated. The formation of ADP from ATP releases part of the C-terminus of KaiC to act as anchor for a KaiA homodimer, which facilitates the release of ADP. Taken together, the experiments performed and the model drawn are a significant progress for the field.

However, I had two questions:

The methods are restricted to one point. Is it therefore feasible to draw a dynamic model? Would it be possible to measure kinetics to monitor binding of KaiA and the kinetics of ADP release before and after? At least the peaks with different ADP content should be quantified.

As per the reviewer's comment, we included the relative quantities in Tables 1 and 2. The reviewer's raises an important issue, which, however, is difficult to address even by the quantification of ADP contents in the ATP-mediated KaiC hexamer. In the revised manuscript, we demonstrated that formation of the AMPPNP-mediated KaiC_{AA} hexamer was hardly affected in the external ATP/ADP condition (Figure S2). In addition, the AMPPNP-mediated KaiC_{AA} hexamer in complex with the KaiA dimer lacked one and two nucleotides (Figure 1C and Table 1). These findings strongly suggested that nucleotides were released after KaiA binding. We modified the sentence to describe these points in the revised manuscript (p.4, lines 8-9).

By comparison of the intensity ratios between figures 5D and 5E it is apparent that the changes occur starting from I497, which is at the border between the A-loop and the C-tail. Hence, is it really possible to state that the A-loop becomes exposed by the interaction with KaiA? It looks like only the C-tail is involved.

We appreciate the comment but are afraid that the reviewer misinterpreted the graphs. The residues involved in the interaction with KaiA are those exhibited peak broadening in Figure 5E, which were distributed not only in the C-tail but also in the A-loop including G488 and I489. On the other hand, the residues highlighted in Figure 5D, i.e. G488, I489, and I497, exhibited an extreme peak broadening due to motional restriction in the AMPPNP-bound state but acquired mobility upon ATP hydrolysis. To avoid the possible confusion, we modified the text with clarity (p.4, lines 26-27).

Minor points:

The figure panel meant in the phrase 'ion peaks (Fig 1C).' is probably Fig 1B.

We would like to apologize for the misnumbering of figure, which has been corrected in the revised manuscript.

Figure legend 1: replace '1 : 1 mixtures' by '1 : 2 mixtures' (or mention KaiA homodimers).

We thank the reviewer for pointing it out. The description was revised in the Figure 1 legend.

Reviewer #3 (Comments to the Authors (Required)):

In this clearly written manuscript by Yunoki and coworkers, "ATP hydrolysis by KaiC promotes its KaiA binding in the cyanobacterial circadian clock system", data is presented demonstrating that ATP hydrolysis enhances the KaiA-KaiC interaction.

The idea that ATP hydrolysis in the C2 ring of KaiC is a prerequisite for KaiA binding is a new contribution to the field, and the data presented here support the authors' contention. The MS and NMR data are of high quality and were interpreted correctly.

Until now, it was thought that the C2 phosphorylation state of KaiC solely dictated the dynamic in/out equilibrium of the A-loops. Whether ADP or ATP were bound by the C2 ring was never considered. Here, data suggest that A-loop exposure requires that ADP occupy at least a fraction of the six nucleotide-binding sites in the C2 ring.

There is one caveat this reviewer would like to point out. The experiments involving ATP were non-physiological in the sense that reaction mixtures contained 100% ATP, rather than more physiological conditions, like an 80%/20% ATP/ADP ratio. Under physiological conditions it is possible that the C2 ring would have ADP at some sites, but not as products of C2 ATP hydrolysis. Thus, it can be envisioned that under physiological conditions the in/out equilibrium of A-loops would be shifted sufficiently toward the out conformation even without a C2 ATPase. This question can be easily addressed by repeating experiments under a physiological range of ATP/ADP ratios. At the very least, however, the authors should discuss how using non-physiological 100% ATP could bias their results.

We thank the reviewer for the constructive comment. We conducted the native MS analysis of the AMPPNP- or ATP-mediated KaiC_{AA} hexamer incubated in the presence of ATP and ADP at varying ratios. The result indicated that the nucleotide states of KaiC_{AA} did not depend on the external ATP/ADP condition. We described these results in the main text by adding Supplementary Figure S2 and in the revised manuscript (p.3, lines 19-21).

NMR peak intensity ratios in Fig. 2D & 2E: any differences in protein concentrations between samples would cause systematic errors. Surprisingly, Fig. 2E suggests that the peak intensity for residue K502 does not change significantly when KaiA binds KaiC. Authors should address how differences in protein concentrations between samples were considered/corrected/eliminated.

We appreciate this useful comment. We carefully checked the peak intensities in the NMR spectra and noticed that the peak originating from K502 was overlapping with those from D514 and E516 in the KaiA-binding condition. Hence, we removed the K502 peak from the spectroscopic probes and revised Figure 2E accordingly.

May 13, 2019

RE: Life Science Alliance Manuscript #LSA-2019-00368-TR

Prof. Koichi Kato
National Institutes of Natural Sciences
Exploratory Research Center on Life and Living Systems (ExCELLS)
5-1 Higashiyama, Myodaiji
Okazaki 444-8787
Japan

Dear Dr. Kato,

Thank you for submitting your revised manuscript entitled "ATP Hydrolysis by KaiC Promotes Its KaiA Binding in the Cyanobacterial Circadian Clock System". As you will see, the reviewers appreciate the introduced changes and reviewer #3 provides constructive input on how to avoid over-interpretation. We would thus be happy to publish your paper in Life Science Alliance pending final revisions:

- please address reviewer #3's concern by appropriate text changes
- please list 10 authors et al. in your reference list
- note that we display suppl figures in-line in the HTML version of the paper => please upload the supplementary figures as individual files and without the legends, the legends can go into the main manuscript docx file

A. FINAL FILES:

- An editable version of the final text (.DOC or .DOCX) is needed for copyediting (no PDFs).
- High-resolution figure, supplementary figure and video files uploaded as individual files: See our detailed guidelines for preparing your production-ready images, <http://www.life-science-alliance.org/authors>
- Summary blurb (enter in submission system): A short text summarizing in a single sentence the

study (max. 200 characters including spaces). This text is used in conjunction with the titles of papers, hence should be informative and complementary to the title. It should describe the context and significance of the findings for a general readership; it should be written in the present tense and refer to the work in the third person. Author names should not be mentioned.

B. MANUSCRIPT ORGANIZATION AND FORMATTING:

Sincerely,

Andrea Leibfried, PhD
Executive Editor
Life Science Alliance
Meyrhofstr. 1
69117 Heidelberg, Germany
t +49 6221 8891 502
e a.leibfried@life-science-alliance.org
www.life-science-alliance.org

Reviewer #1 (Comments to the Authors (Required)):

The authors have completely addressed the major and minor concerns from the first round of review. This is a thoughtful study that addresses an important, mechanistic step in the cyanobacterial circadian oscillator and, therefore, should be of interest to the field of circadian biology.

Reviewer #2 (Comments to the Authors (Required)):

The authors have addressed my questions. Overall, the manuscript improved and the revisions are well incorporated. I just found this sentence, in which the verb and explanation of the impact of the mutation seem to lack: Moreover, we examined KaiA binding to a KaiCAA mutant in which a catalytic glutamate (Glu77) in the CI ATPase domain (designated as KaiCAA/E77Q), indicating that this mutation affected neither the nucleotide state nor the KaiA-binding property of the KaiCAA hexamer (Fig S4 and Table 2).

Reviewer #3 (Comments to the Authors (Required)):

As with the original manuscript in this revision, the authors provide evidence that an ATPase activity in the CII ring of KaiC is necessary to promote A-loop exposure and hence KaiA-KaiC interactions that promote nucleotide exchange of ADP for ATP. Although the data are supportive of this new model, this reviewer wants to add a caveat. In an effort to address the role of the CI ATPase in KaiA-KaiC interactions, the authors showed that the E77Q mutant of KaiC-AA binds KaiA just as well as KaiC-AA without the mutation. The authors state that because the E77Q KaiC-AA mutant does not have CI ATPase activity, it is not needed for KaiA-KaiC binding. However, the literature suggests that E78 is also a catalytic glutamyl residue in CI (see Phong et al, 2013, PNAS, 110:1124-1129). Because the authors of this revised manuscript did not compare the ATPase activities of KaiC-AA, E77Q KaiC-AA, and E77Q/E78Q KaiC-AA, this reviewer thinks that it is premature to claim that the E77Q mutation alone abolishes CI ATPase activity. The authors should address this in their next revision.

May 17, 2019

RE: Life Science Alliance Manuscript #LSA-2019-00368-TRR

Prof. Koichi Kato
National Institutes of Natural Sciences
Exploratory Research Center on Life and Living Systems (ExCELLS)
5-1 Higashiyama, Myodaiji
Okazaki 444-8787
Japan

Dear Dr. Kato,

Thank you for submitting your Research Article entitled "ATP Hydrolysis by KaiC Promotes Its KaiA Binding in the Cyanobacterial Circadian Clock System". It is a pleasure to let you know that your manuscript is now accepted for publication in Life Science Alliance. Congratulations on this interesting work.

DISTRIBUTION OF MATERIALS:

Again, congratulations on a very nice paper. I hope you found the review process to be constructive and are pleased with how the manuscript was handled editorially. We look forward to future exciting submissions from your lab.

Sincerely,

Andrea Leibfried, PhD
Executive Editor
Life Science Alliance
Meyerohofstr. 1
69117 Heidelberg, Germany
t +49 6221 8891 502
e a.leibfried@life-science-alliance.org
www.life-science-alliance.org